# Bioconversion of Glycerol to 1,3-Propanediol Using *Klebsiella pneumoniae* L17 with the Microbially Influenced Corrosion of Zero-Valent Iron

Da Seul Kong [1], Minsoo Kim [1], Shuewi Li [1], Sakuntala Mutyala [1], Min Jang [2], Changman Kim [3,4,*] and Jung Rae Kim [1,*]

[1]  School of Chemical Engineering, Pusan National University, Busan 46241, Republic of Korea
[2]  Department of Environmental Engineering, Kwangwoon University, 20 Kwangwoon-Ro, Nowon-Gu, Seoul 01897, Republic of Korea
[3]  Advanced Biofuel and Bioproducts Process Development Unit, Lawrence Berkeley National Laboratory, Emeryville, CA 94608, USA
[4]  Department of Biotechnology and Bioengineering, Chonnam National University, Gwangju 61188, Republic of Korea
*  Correspondence: cmkim@jnu.ac.kr (C.K.); j.kim@pusan.ac.kr (J.R.K.); Tel.: +1-510-332-0104 (C.K.); +82-51-510-2393 (J.R.K.); Fax: +82-51-512-8563 (C.K. & J.R.K.)

**Abstract:** The bacterial redox state is essential for controlling the titer and yield of the final metabolites in most bioconversion processes. Glycerol conversion to 1,3-propanediol (PDO) requires a large amount of reducing equivalent and the expression of reductive pathways. Zero-valent iron (ZVI) was used in the glycerol bioconversion of *Klebsiella pneumoniae* L17. The level of 1,3-PDO production increased with the oxidation of ZVI ($31.8 \pm 1.2$ vs. $25.7 \pm 0.5$, ZVI vs. no ZVI) while the cellular NADH/NAD$^+$ level increased (0.6 vs. 0.3, ZVI vs. no ZVI). X-ray diffraction showed that the iron oxide ($Fe_2O_3$) was formed during glycerol fermentation. L17 obtained electrons from ZVI and dissolved the iron continuously to form cracks on the surface, suggesting microbially influenced corrosion (MIC) was involved on the surface of ZVI. The ZVI-implemented fermentation shifted bioconversion to a more glycerol-reductive pathway. The qPCR-presented glycerol dehydratase (DhaB) with ZVI implementation was strongly expressed compared to the control. These results suggest that ZVI can contribute to the biotransformation of PDO by inducing intracellular metabolic shifts. This study could also suggest a novel microbial fermentation strategy with the application of MIC.

**Keywords:** metabolic shift; zero-valent iron (ZVI); iron-supported bioconversion; *Klebsiella pneumoniae* L17′ 1,3-propanediol(1,3-PDO); glycerol



## 1. Introduction

The cost of steel destruction and replacement induced by corrosion accounts for approximately 3% of global gross domestic product [1,2]. Microbe–metal interactions are an essential ecological niche in the earth's biogeochemical cycles for metal and metalloid compounds. The worldwide cost of microbially influenced corrosion (MIC) is reaching billions of dollars per year [3,4]. Various microorganisms involved in MIC have been reported to be iron-oxidizing (IOB), sulfate-reducing (SRB), nitrate-reducing (NRB), acid-producing (APB), and metal-oxidizing bacteria (MOB), as well as methanogens, fungi, and even archaea [5–8]. MIC can transfer respiratory electrons anaerobically from the metallic iron surface to a cell, where energetically favorable redox reactions proceed and provide respirational energy for the cellular metabolism, called electrical MIC (EMIC) [3,9–12]. IOB oxidizes iron species using organic matter (e.g., glycerol) to synthesize biomass [13]. IOBs frequently form a gelatinous or fibrous outer layer that provides a favorable anaerobic environment such as a micro-electrochemical cell [14]. Electrons from iron oxidation are

delivered to iron oxidase or c-type cytochromes on the outer membrane and reduce $NAD^+$ to NADH by inner membrane NADH dehydrogenase [15–17]. Other reports showed that such iron oxidation shifts the bacterial metabolic pathway and influences the intracellular redox state [18–21].

1,3-Propanediol (1,3-PDO; $C_3H_8O_2$) is an important intermediate in the synthesis of polymers and value-added chemicals needed to replace petrochemical-based platform compounds [22,23]. 1,3-PDO is used widely in the industrial production of polyester polytrimethylene terephthalate (PTT) [24,25]. The biosynthesis of 1,3-PDO uses the glycerol-reductive pathway via 3-HPA (Equations (1) and (2)) [26,27]. The continuous reproduction of NADH is essential for the efficient production of PDO [28–30]. Various strategies for regenerating NADH from $NAD^+$ for PDO production have been reported [19,30]. *K. pneumoniae* is used as a biocatalyst for glycerol conversion to synthesize glycerol derivatives, such as 1,3-PDO and 3-HP, owing to its highly developed anaerobic glycerol assimilation capability. Among *K. pneumoniae* strains, *K. pneumoniae* L17 can transfer their residual electron to extracellular metallic compounds (or carbon electrode) [24,28,31,32]. Electro-fermentation could control the bacterial redox balance to increase the productivity of value-added chemicals in bioconversion [18,32–35]. On the other hand, MIC-driven bioconversion and fermentation have not been studied extensively. The concept of MIC can be applied easily to bioconversion by adding powder or granule-type metal compounds to a bioreactor.

$$\text{Glycerol}(C_3H_8O_3) \rightarrow \text{3-HPA } (C_3H_6O_2) + H_2O \tag{1}$$

$$\text{3-HPA}(C_3H_6O_2) + 2\text{NADH} \rightarrow \text{1,3-PDO}(C_3H_8O_2) + 2\text{NAD}^+ \tag{2}$$

This study evaluated the bioconversion of glycerol to PDO using zero-valent iron (ZVI) as a reducing reagent. Previous studies optimized fermentation with ZVI in minimal media without a vitamin and trace elemental solution under various conditions (e.g., oxygen and pH) [28,36]. The present study examined the mechanism of the cellular metabolism and iron ion dynamics ($Fe^{2+}$ and $Fe^{3+}$ from ZVI oxidation) during MIC. The bacterial metabolic shift induced by ZVI was first evaluated. The transcription level and metabolite-based mathematic model were investigated in *Klebsiella pneumoniae* L17. The changes in dissolved ferric ($Fe^{3+}$) and ferrous ($Fe^{2+}$) ions and ZVI surface analysis indicated the dynamics of iron species in glycerol conversion. These results contribute to the development of a novel microbial conversion and fermentation by applying MIC.

## 2. Materials and Method

### 2.1. Strain and Cultivation

The source of *K. pneumoniae* L17 was the China Center for Type Culture Collection (CCTCC). Luria Bertani (LB) medium was used for the preculture and seed culture of *K. pneumoniae* L17. The fermentation medium for flask experiments has a composition of 18.04 g $K_2HPO_4$, 1.804 g $KH_2PO_4$, 1 g NaCl, 1 g $NH_4Cl$, 0.2 g yeast extract, 0.25 g $MgSO_4 \cdot 7H_2O$, 9.2 g pure glycerol, 1.25 mL $100\times$ vitamin solution (per liter: biotin 2 mg, folic acid 2 mg, pyridoxine hydrochloride 10 mg, thiamine hydrochloride 5 mg, nicotinic acid 5 mg, D, L-calcium pantothenate 5 mg, vitamin $B_{12}$ 0.1 mg, p-aminobenzoic acid 5 mg, and lipoic acid 5 mg) and 1.25 mL $100\times$ trace element solution (per liter: nitrilotriacetic acid 1 g, $MgSO_4 \cdot 7H_2O$ 3 g, $MnSO_4 \cdot 2H_2O$ 0.5 g, NaCl 1 g, $FeSO_4 \cdot 7H_2O$ 0.1 g, $CoSO_4$ 0.1 g, $CaCl_2$ 0.1 g, $CuSO_4 \cdot 5H_2O$ 0.01 g, $Al \cdot K(SO_4)_2$ 0.01 g, $H_3BO_3$ 0.01 g, $NiCl_2$ 0.024 g, $Na_2WO_4$ 0.025 g, $ZnCl_2$ 0.13 g, and $Na_2MoO_4 \cdot 2H_2O$ 0.025 g). HCl (10%) and NaOH (5 N) solutions were used to adjust the pH of the medium to 7.5. ZVI (0.5 g, 70 mesh powder, <212 micrometer, ACROS, New Jersey, NJ, USA), ferric chloride (0.2 g, $FeCl_3$, Duksan, Ansansi, gyunggido, Korea), and ferrous chloride tetrahydrate (0.31 g, $FeCl_2 \cdot 4H_2O$, Fisher Chemical, Waltham, MA, USA) were added as an external iron supplement as an electron donor. *K. pneumoniae* L17 (hereinafter L17) was cultured in a shaking incubator at 200 rpm and 30 °C. Anaerobic cultivation was carried out after nitrogen purging into the headspace of a cultivation serum bottle (200 mL) for 10 min. The initial cell density of the culture was adjusted to 0.05 $OD_{600}$.

### 2.2. Analytical Methods

The liquid samples were taken at 0, 5, 10, and 24 h to determine the cell density and metabolite production. The cell density ($OD_{600}$) was measured using a UV-Vis spectrophotometer (Optizen POP, Keen Innovative Solutions, Daejeon, Korea). Metabolite production was identified by centrifuging each liquid sample at 5000 rpm for 10 min, diluting the supernatants, and filtering them through a syringe filter. The samples were analyzed by high-performance liquid chromatography (HPLC, HP 1160 series, Agilent Technologies, Santa Clara, CA, USA) equipped with a $300 \times 7.8$ mm Aminex HPX-87H (Bio-Rad, Hercules, CA, USA) column at 65 °C and a refractive index (RI) and photodiode array (PDA) detector, using 2.5 mM $H_2SO_4$ as the mobile phase (flow rate = 0.5 mL/min). The concentrations of total dissolved iron and ferrous ions ($Fe^{2+}$) were quantified using colorimetric methods [28]. The iron species in solution were measured using a Fe ion kit (HS-Fe(T), HUMAS, Daejeon, Korea) and $Fe^{2+}$ ions kit (HS-Fe (2+), HUMAS, Daejeon, Korea). $Fe^{3+}$ was calculated by subtracting the measured $Fe^{2+}$ from the total iron concentration. Powder X-ray diffraction (XRD, X'Pert-MPD, PHILIPS, Amsterdam, Netherlands) was performed using Cu-K$\alpha$ ($\lambda$ = 0.15406 nm) operated at 40 kV and 30 mA. Metabolic flux analysis (MFA) was investigated, as described elsewhere [37]. All the equations for the undetermined metabolic pathway were solved by linear optimization using the MetaFluxNet program [38]. $NADH/NAD^+$ ratio was measured using a colorimetric kit (K337-100, Biovision, Milpitas, CA, USA) [18].

### 2.3. Real-Time qPCR

The cells were harvested from late-exponential growth phases to analyze the expression level of each gene in the glycerol fermentation pathway. The culture was centrifuged at 8000 rpm for two minutes. The harvested pellet was processed for RNA isolation using a total isolation kit (TransZol UP, TRAN, Beijing, China). RNA was measured and calculated by nano-drop (NanoDrop One$^c$, Thermo Scientific, Waltham, MA, USA) for the same sample concentration. The estimated amount of RNA was synthesized into cDNA using the first-strand cDNA synthesis system (iSccript cDNA synthesis kit, BIO-RAD, Hercules, CA, USA). qPCR analysis was performed using the SYBR green method in the StepOnePlus Real-Time PCR system (Applied Biosystems, Waltham, MA, USA). Each 20 μL sample of reaction mixture contained 300 ng of cDNA, 10 μL of 2xPower SYBR Green PCR Master Mix (Applied Biosystems, Waltham, MA, USA), 5 pmol of forward and reverse primers, and DEPC-treated water. The thermal cycling conditions were as follows: one cycle of denaturation at 95 °C for 30 s; 40 cycles of amplification at 95 °C for 15 s, 62 °C for 30 s, and 72 °C for 30 s. The relative quantification of the mRNA levels was calculated using the $\Delta\Delta C_t$ method [36]. All analyses were carried out in duplicate.

## 3. Results and Discussion

### 3.1. Metabolic Profile of K. pneumoniae L17 with ZVI

Three different iron species (ZVI ($Fe^0$), $Fe^{2+}$, and $Fe^{3+}$) were added for anaerobic glycerol fermentation. Most of the glycerol was consumed in all bottles ($104.3 \pm 1.2$, $102.3 \pm 0.2$, and $102.8 \pm 0.1$ for glycerol consumption in the control, ZVI, ferrous and ferric, respectively) within 24 h. Consequently, PDOs were produced to $25.7 \pm 0.5$, $31.8 \pm 1.2$, $27.5 \pm 0.4$, and $25.8 \pm 0.1$ mM in the control, ZVI, ferrous ($Fe^{2+}$), and ferric ($Fe^{3+}$), respectively. The yield of PDO production was 1.3 times higher with ZVI (0.31 PDO production mole/glycerol consumption mole) than with the control (0.24). Figure 1 and Table S1 present the metabolite productions. Interestingly, the metabolic profiles of L17 with more oxidized iron species ($Fe^{2+}$ and $Fe^{3+}$) showed similar glycerol consumption and PDO production to the control. This indicates that L17 can uptake electrons from a solid granule of more reduced iron (ZVI) rather than the oxidized form of dissolved $Fe^{2+}$ and $Fe^{3+}$. The metabolic flux model was suggested to estimate the fluxes in bioconversion and the microbial redox states of the oxidative and reductive pathways (Figure 2). ZVI shifted the metabolic flux toward more reductive pathways (30.6 and 46.9 in the control and ZVI, respectively). Kim et al.

reported that under cathodic glycerol fermentation, electrode-based electron transfer drives the metabolic shift of L17 from 30:70 (reductive to oxidative pathways) under open circuit conditions (non-BES) to 40:60 under cathodic conditions (BES) [24]. In this study, only 0.23 mmol of electron transfer resulted in 6.1 mmol of PDO increase. Here, ZVI played a similar role to the electrode to provide electrons to the cell continuously while inducing a shift in metabolic flux (Figure 2).

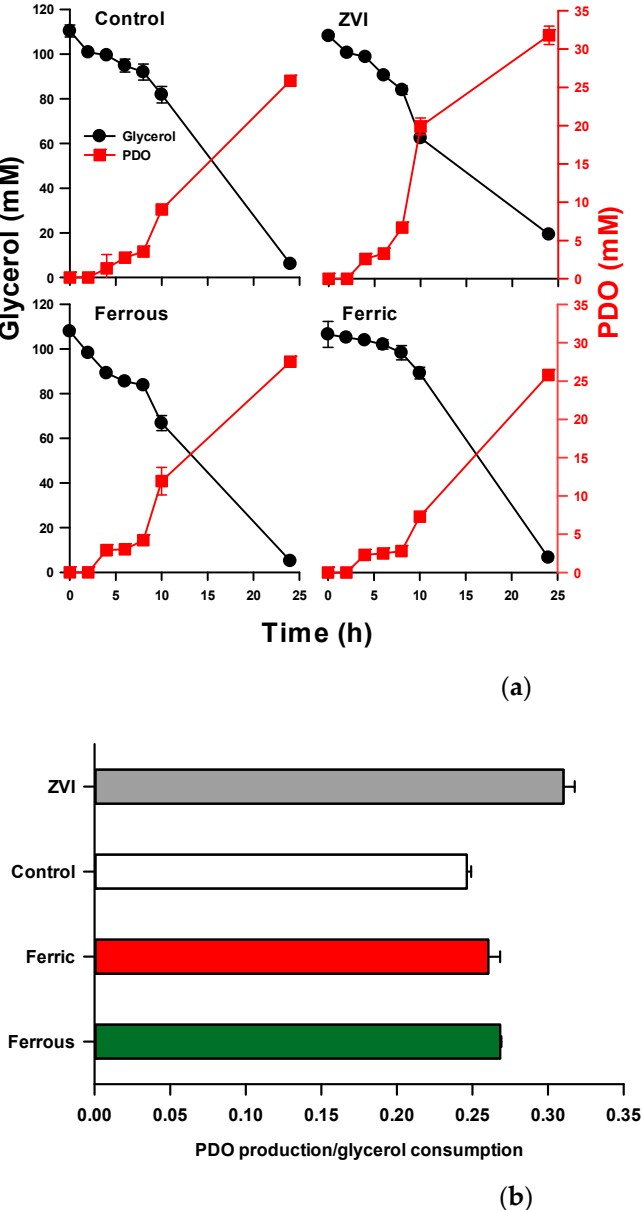

(**a**)

(**b**)

**Figure 1.** Metabolite production of *K. pneumoniae* L17 using iron species (ZVI, ferrous, ferric, without iron). (**a**) Glycerol consumption and PDO production; (**b**) 1,3-PDO production yield with iron species (ZVI, ferrous, ferric, and control). ZVI was used as an electron donor for bacteria, while the control did not contain ZVI. Ferric ($Fe^{3+}$) in the form of $FeCl_3$ ($Fe^{2+}$) and ferrous in the form of $FeCl_2 \cdot 4H_2O$ were used as alternative iron species to ZVI ($Fe^0$).

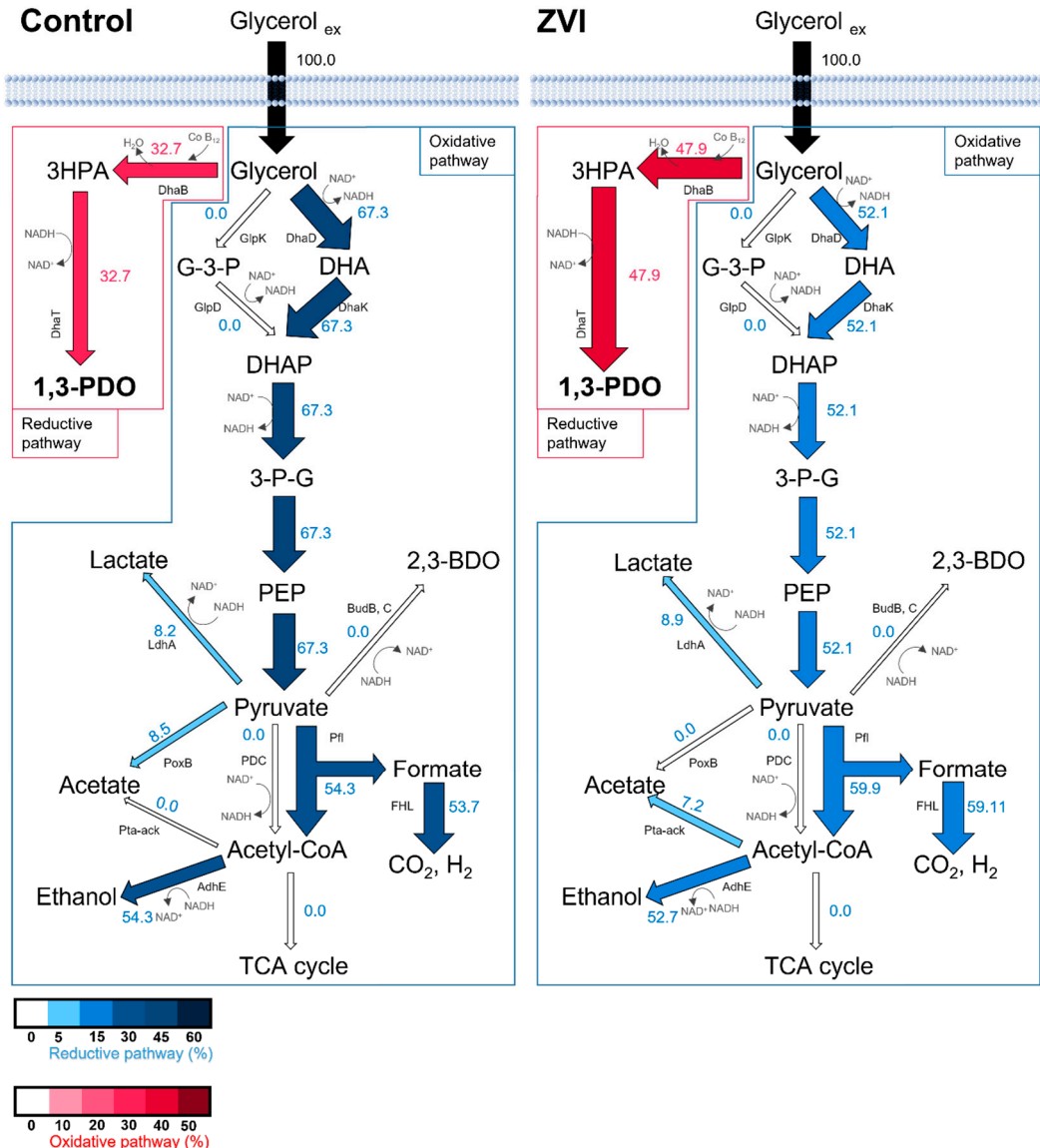

**Figure 2.** Metabolic flux analysis with and without ZVI, which was calculated using the meta flux net program. (Control) without ZVI and (ZVI) with ZVI in the reactor. Glycerol_ex indicates that glycerol existed outside of the cell membrane.

### 3.2. Change of Bacterial Redox State by ZVI

The bacterial redox state is influenced by external reducing and oxidizing reagents in glycerol bioconversion. A previous study showed that the $NADH/NAD^+$ ratio, representing the cellular redox state, was controlled by the potential applied to the electrode [37]. Similar to previous work, Figure 3 showed that the $NADH/NAD^+$ ratio with ZVI was relatively higher than that in the control. The intracellular electron transfer of L17 under ZVI maintained a high $NADH/NAD^+$ level throughout fermentation. A more reduced environment shifted glycerol conversion toward the reductive pathway. Under anaerobic conditions, L17 converts glycerol to two alcoholic metabolites, PDO and ethanol, by alcohol dehydrogenases with NADH as a cofactor, which is more favorable under high cytoplasmic $NADH/NAD^+$ ratios. The net NADH changes in PDO and ethanol production were $-1$ and $0$, respectively; PDO production is preferred to reduce the cytoplasmic NADH level. When additional reducing energy was added to the cytoplasm by ZVI oxidation, the cytoplasmic $NADH/NAD^+$ ratio would increase, and the metabolic fluxes would shift to maintain the cellular redox homeostasis. PDO production would decrease the

cytoplasmic NADH/NAD$^+$ ratio with the concomitant regeneration of NADH. Although the increase in PDO production in ZVI-implemented fermentation was identified, a high NADH/NAD$^+$ ratio was maintained compared to the control. This result suggests the following: (1) the steady supplement of electrons into L17 or (2) insufficient catalytic activity of the glycerol-reductive pathway to resolve the high NADH/NAD$^+$ ratio.

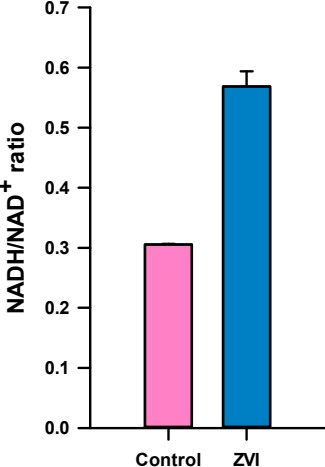

**Figure 3.** NADH/NAD$^+$ ratio with and without ZVI. (Control) without iron in the reactor and (ZVI) with ZVI in the reactor.

The redox-related metabolic response to maintain redox homeostasis is usually initiated by the transcription level. L17 has three major modules (enzymes) to initiate glycerol conversion depending on the cell conditions: aerobic (via GlpK), anaerobic oxidation (via DhaD), and anaerobic reduction (via DhaB). The transcriptional analysis of such enzymes was carried out to reveal the metabolic response against ZVI implementation. Figure 4 presents the comparative mRNA level of the three enzymes. Interestingly, the expression of glycerol dehydratase (dhaB), the first enzyme of glycerol reduction, was approximately three times higher with ZVI than with the control.

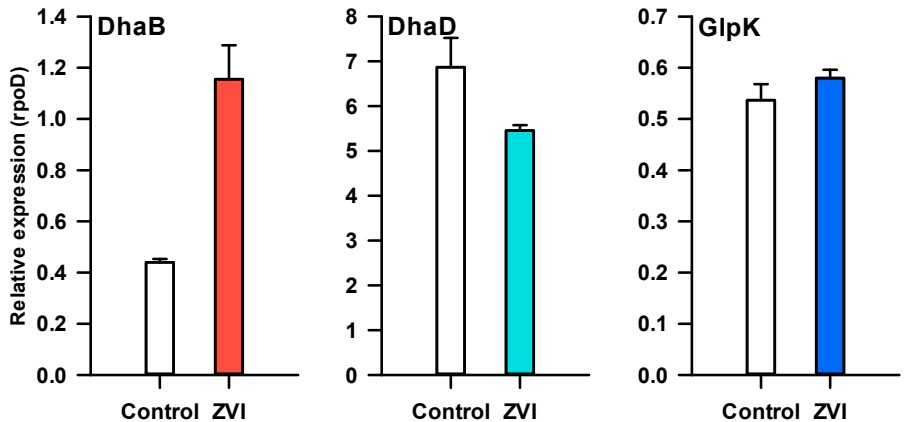

**Figure 4.** mRNA expression profiles for glycerol-converting enzymes (glpK, dhaD, and dhaB) with ZVI and without ZVI. (Control) without iron in the reactor and (ZVI) with ZVI in the reactor.

In contrast, glycerol dehydrogenase (dhaD) for anaerobic glycerol oxidation was less expressed with ZVI than the control. This result suggests that ZVI-driven electron transfer shifts the conversion toward the reductive metabolism. Although the glycerol-reductive pathways are well expressed with ZVI, the NADH/NAD$^+$ ratio was still high (Figure 3). Therefore, PDO production was insufficient to release the intracellular redox

stress induced by ZVI. The glycerol consumption and PDO production were similar under higher coenzyme $B_{12}$ levels, an essential cofactor for glycerol bioconversion (Figure S1).

The ZVI and control cultures expressed similar glycerol kinase (glpK) levels. The glpK phosphorylates glycerol to synthesize glycerol-3-phosphate (G-3-P) in the glp module. G-3-P is oxidized further by glycerol-3-phosphate dehydrogenase (glpD) to synthesize dihydroxyacetone-phosphate (DHAP). GlpR (glycerol-3-phosphate regulon repressor) inhibits the expression of glp regulon. GlpR has a high affinity with G-3-P, and the complex formation of G-3-P and GlpR prevents the expression inhibition of the glp module enzymes (glpK and glpD) [39]. The Glp module functions under aerobic and anaerobic respiratory conditions [40]. Therefore, anaerobic and additional reducing supplement conditions, both ZVI and an electrode, did not influence glpK expression significantly.

### 3.3. Oxidation of ZVI by Microbially Influenced Corrosion (MIC)

Figures 5 and S2 show the change in soluble iron species in the culture medium during the experiment. When ZVI was implemented in L17 (biotic), the level of $Fe^{2+}$ increased over 10 h, whereas it did not increase in the abiotic and inactive cultures (Figure 5). This result indicates that the bacterial activity oxidized ZVI for their metabolism. On the other hand, the dissolved iron in the media decreased regardless of the cell throughout cultivation, probably due to the precipitation of $Fe_2O_3$ or $Fe_3O_4$ (Figures 5 and S2). In a separate experiment, when ZVI was autoclaved for sterilization, the pH increased slightly from 7.5 to 8.2. This result can be attributed to the substances inside the culture medium reacting with ZVI at high temperatures; ZVI is oxidized (Equations (3) or (4)), and water is decomposed (Equation (5)) to form iron oxide ($Fe_3O_4$ or $Fe_2O_3$). The generated $OH^-$ increases the pH of the culture medium with the accumulation of $H_2$ in the headspace (Equations (6) or (7)) (Figure S3) [41–43].

$$Fe \rightleftharpoons Fe^{2+} + 2e^- \tag{3}$$

$$Fe \rightleftharpoons Fe^{3+} + 3e^- \tag{4}$$

$$H_2O \rightleftharpoons OH^- + H^+ \tag{5}$$

$$Fe^{2+} + 2Fe^{3+} + 6OH^- + 6H^+ + 8e^- \rightleftharpoons Fe_3O_4 + 2OH^- + 5H_2 \uparrow \tag{6}$$

$$2Fe^{3+} + 5OH^- + 5H^+ + 6e^- \rightleftharpoons Fe_2O_3 + 2OH^- + 4H_2 \uparrow \tag{7}$$

XRD was carried out on the iron oxide precipitates on the bottom of the culture: abiotic ZVI with non-sterilization (ab-ZVI-ns); abiotic ZVI with sterilization (ab-ZVI-s); ZVI after 24 h culture without bacteria (ab-ZVI-24h), and ZVI after 24 h culture with bacteria (bio-ZVI-24h) [44]. The peak intensity for $Fe_2O_3$ was lower in ab-ZVI-ns, while it was higher in bio-ZVI-24h (Figure 6), but the peak for $Fe_3O_4$ was negligible (Figure S4). $Fe^{2+}$ was oxidized further to $Fe^{3+}$ and precipitated in the form of $Fe_2O_3$. With the other form of flat-type ZVI specimen, cracks were found from 10 and 24 h of cultivation with L17 (Figure 7). L17 dissolves iron continuously from the ZVI surface by oxidation and eventually fractures the specimen. This crack formed under the anaerobic environment. The first hypothesis is that L17 attaches to the surface of the ZVI specimen to exchange electrons by oxidation and then detaches from the surface (Figure 8A) [45,46]. The second is that MIC is initiated by bacterial attachment (cathode site), which continuously drives electron flow from the neighboring anode site [9,47,48]. The continuous oxidation of iron by MIC eventually forms a linear crack, as shown in Figure 8B. More cracks formed at bio-ZVI-24h than at 10 h (Figure 7).

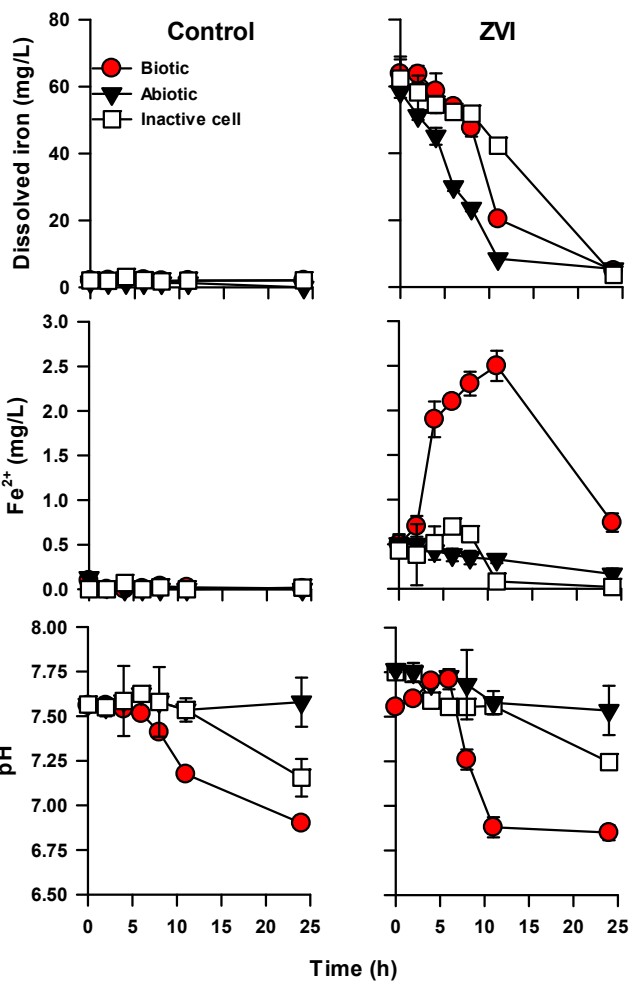

**Figure 5.** Changes in the iron concentration (dissolved iron and $Fe^{2+}$) and pH for the control (without iron), and with ZVI during glycerol fermentation.

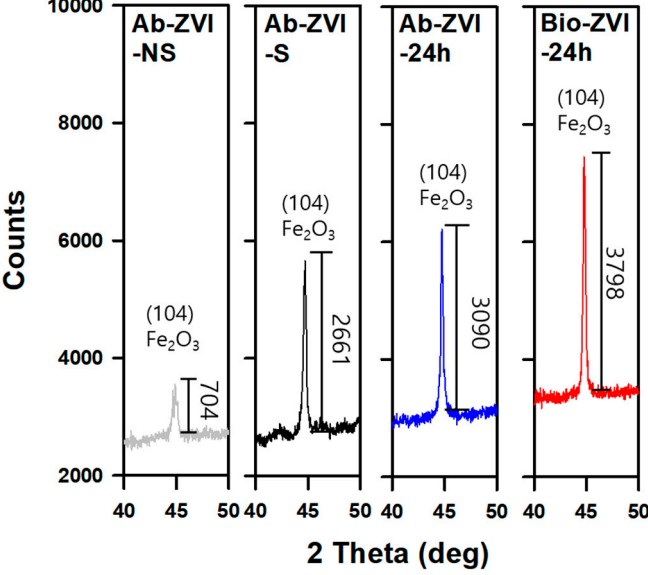

**Figure 6.** XRD pattern of iron precipitation of ab-ZVI-ns, ab-ZVI-s, ab-ZVI-24h, and bio-ZVI-24h.

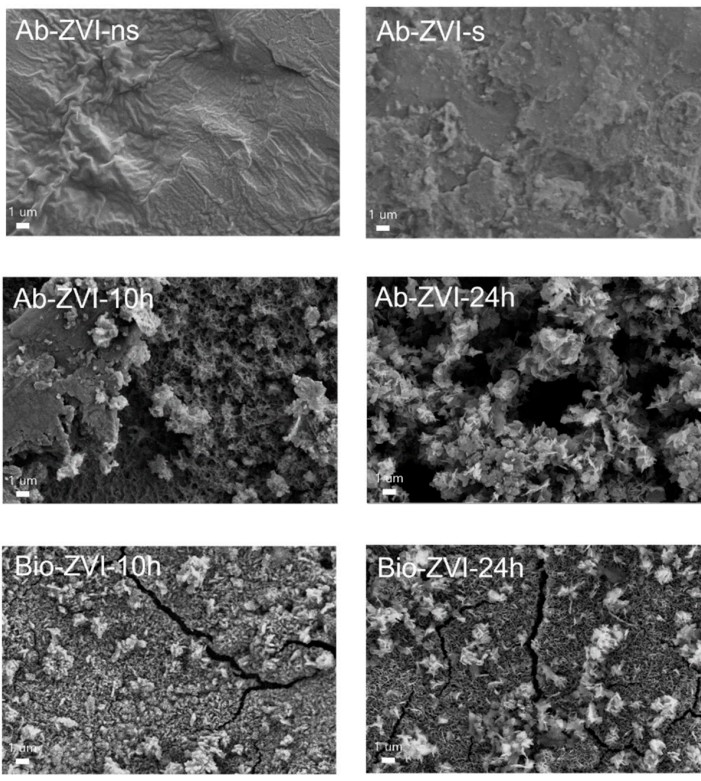

**Figure 7.** FE-SEM images of the surface of the ZVI specimen before the experiment, ZVI after sterilization, and during the abiotic and biotic culture for 10 h and 24 h.

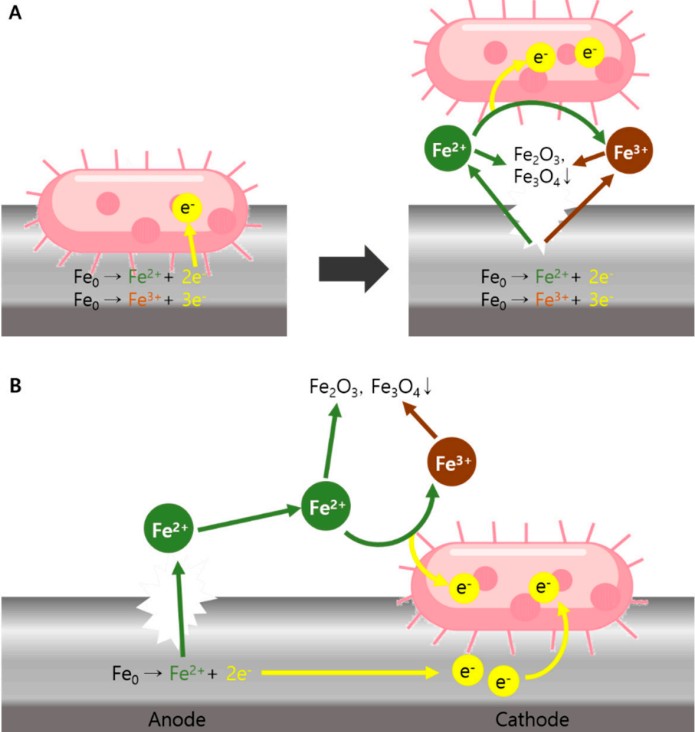

**Figure 8.** Schematic diagram of microbiologically influenced corrosion (MIC) for L17 at ZVI. (**A**) cell attaches to the surface of the ZVI specimen to exchange electrons by oxidation and then detaches from the surface. (**B**) Microbially influenced corrosion (MIC) is initiated by bacterial attachment (cathode site), which continuously drives electron flow from the neighboring anode site.

Microbially influenced corrosion (MIC) of ZVI provide electrons to activate the glycerol-reductive pathway and simultaneously increase 1,3-PDO production. These results showed that *Klebsiella pneumoniae* L17, which was previously known as a glycerol fermenter, can uptake respiratory electrons from MIC and drive bioconversion to a more reductive pathway. The MIC with bioconversion changes the surface morphology of ZVI. Although L17 was reported to be active in using carbon electrodes as the electron acceptor and donor in a bioelectrochemical system, it could also oxidize iron species under an anaerobic environment in this study. Hence, the corrosion proceeds by a wider range of bacterial species, including fermentative strains over the conventionally known iron-oxidizing bacteria. The ZVI has been studied extensively as a reducing reagent in environmental applications. Thus, it can also be applied to bioconversion and fermentation. Value-added chemicals can be produced by providing cost-effective iron powder, such as ZVI, in fermenters and bioprocesses. The cost of iron powder (ZVI) is approximated at USD 7.6/kg with considering the higher market price of 1,3-PDO (USD 194/kg). Only 0.5 g per flask (approx. 0.38 cent) was implemented for general culture in this study. Although further development of scaled-up fermenter and process should be carried out, this work has shown the feasibility of ZVI application to glycerol conversion.

## 4. Conclusions

This study examined the cellular metabolism and dynamics of ZVI oxidation during glycerol conversion by *Klebsiella pneumonia* L17. L17 receives reductive energy from ZVI oxidation to increase the NADH level within cells. The NADH/NAD$^+$ ratio changes from 0.3 (control) to 0.6 (with ZVI) when ZVI is provided, and L17 prefers the reductive pathway to consume NADH (32.7%:47.9%, control: with ZVI, metabolic flux). Therefore, the increase in glycerol dehydratase (DhaB) was observed in ZVI compared to the control using qPCR. In particular, L17 oxidizes ZVI and forms cracks on the ZVI surface. XRD revealed $Fe_2O_3$ to be the solid iron species in the bottle. These results suggest that reductive iron species, such as ZVI, induce an intracellular metabolic shift in bioconversion. These results show that the microbially influenced corrosion (MIC)-driven bioconversion helps control the redox balance in microbial fermentation. This study is also expected to develop a novel microbial fermentation strategy using cost-effective reducing agents, such as ZVI.

**Supplementary Materials:** The following supporting information can be downloaded at: https://www.mdpi.com/article/10.3390/fermentation9030233/s1, Table S1: Metabolite production of *K. pneumoniae* L17 using iron species (ZVI, ferrous, ferric, and without iron); Figure S1: PDO productivity and glycerol consumption with various coenzyme $B_{12}$ concentrations in the medium. (A) Glycerol consumption and (B) 1,3-PDO production; Figure S2: Change in the concentration of iron species and pH of *K. pneumoniae* L17 with initial ferric and ferrous ions; Figure S3: Change in the headspace gas concentration (change in hydrogen and nitrogen) before (ab-ZVI-ns) and after autoclave (ab-ZVI-s); Figure S4: XRD patterns of ab-ZVI-ns, ab-ZVI-s, ab-ZVI-24h, and bio-ZVI-24h ($Fe_3O_4$).

**Author Contributions:** Conceptualization, C.K.; Methodology, D.S.K., S.L. and S.M.; Validation, M.K. and S.M.; Formal analysis, S.L.; Investigation, D.S.K.; Data curation, M.K. and S.M.; Writing—original draft, D.S.K.; Writing— review and editing, M.J., C.K. and J.R.K.; Supervision, J.R.K.; Funding acquisition, J.R.K. All authors have read and agreed to the published version of the manuscript.

**Funding:** This study was supported by the Mid-Career Researcher Program (NRF-2021R1A2C200784111), the Basic Research Program (NRF-2022R1F1A1063317) through the National Research Foundation of Korea (NRF), and BK21 FOUR Program by Pusan National University Research Grant, 2021. This work was also supported by the Basic Research Laboratory Program (BRL) (NRF-2022R1A4A1021692) by the NRF, funded by the Korean Ministry of Science, ICT, and Future Planning.

**Institutional Review Board Statement:** Not applicable.

**Informed Consent Statement:** Not applicable.

**Data Availability Statement:** Data will be made available on request.

**Conflicts of Interest:** The authors declare no conflict of interest.

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
