# Peer review of "Bioconversion of Glycerol to 1,3-Propanediol Using Klebsiella pneumoniae L17 with the Microbially Influenced Corrosion of Zero-Valent Iron"

_fermentation, doi:10.3390/fermentation9030233_

Round 1
Reviewer 1 Report
Bioconversion of glycerol to 1,3-propanediol using Klebsiella
pneumoniae L17 with the microbially influenced corrosion of
zero-valent iron
#### General comments:
The reference 28 from the same laboratory as the authors of this mansucript
seem to contain essentially the same experiments and results as the present
manuscript. Some phrases in the new manuscript are exactly the same, like:
"K. pneumoniae L17 was purchased from the China Center for Type Culture Collection
(CCTCC)." Which occur in both manuscripts.
The authors should refer to their previous work on L64-67 in amore specific way. The
authors need to be more specific than simply claiming that "iron ion dynamics"
was measured. What scientific question arose in 28 that they are now testing?
Figure 1 show a final difference of titer between 1,3-propanediol for control and ZVI
of about 5 mM. This translates to 5 mM NADH needed for the reduction
Figure 5 shows Fe2+ being produced peaking at about 2.5 mg/L at about 10 h on
the middle right panel. This translates to about 0.045 mM electron pairs (or NADH equivalents)
produced.
2.5 mg/L = 1000 * (2.5/1000)/55.845 = 0.045 mM Fe 2+ <=> 0.045 * 2 = 0.09 mM e-
Hence, the observed production of Fe 2+ in Figure 5 can only produce about 1/100th of the needed
extra reducing power for the higher titer observed in Figure 1.
The authors need to discuss this in a future version of this manuscript.
The English is good throughout the mansucript.
(28) D.S. Kong, C. Kim, Y.E. Song, J. Baek, H.S. Im, J.R. Kim, Zero-valent iron
driven bioconversion of glycerol to 1, 3-propanediol using Klebsiella
pneumoniae L17, J Process Biochemistry energies (2021).
#### Specific comments:
Supplementary data, on the first page: Republic of Korae => Republic of Korea
L79 Specify content of vitamin solution and trace (element) solution.
L79 trace solution -> trace element solution
L104 Meta Flux Net -> MetaFluxNet
Author Response
We have appropriately response to reviewer's comment and concern in the attached file

Reviewer 2 Report
The application of MIC for PDO production is interesting, and the manuscript is relatively well-written. Here are my comments for the authors to consider during revision: Describe the cost of using MIC: Is it economically feasible to apply this strategy for industrial-scale fermentation? Is Klebsiella pneumonia an (opportunistic) pathogenic bacterium and can it be used for industrial production? Whether MIC and Klebsiella pneumonia have been used in industrial applications?
In addition, the figures can be reorganized and made more professional (e.g., Figure 1a, b can be in the same horizontal orientation, same for figure 5). In addition, provide more details in the legend of the figures. The method for NADH/NAD+ ratio determination should be provided. The format of reference also needs improvement.
Author Response

(The authors gave the same response as above.)
